# Simulation Analysis and Performance Test of a Compressible Piezoelectric Pump Fluid Cavity

**DOI:** 10.3390/mi13071100

**Published:** 2022-07-13

**Authors:** Xia Liu, Tongyu Wang, Hu Wang, Jun Hou, Jinlong Liu, Jiaying Lin, Shenfang Li, Zhicong Wang, Xiaochao Tian, Zhigang Yang

**Affiliations:** 1School of Mechatronical Engineering, Changchun University of Science and Technology, Changchun 130022, China; ccdxliuxia@163.com; 2School of Mechanical and Vehicle Engineering, Changchun University, Changchun 130022, China; wanghuccdx@163.com (H.W.); hj18843258090@163.com (J.H.); a1971253187@163.com (J.L.); jiaying_lin_sr@163.com (J.L.); lsfdyx0408@163.com (S.L.); w18844010813@163.com (Z.W.); tianxczb@163.com (X.T.); 3College of Mechanical and Aerospace Engineering, Jilin University, Changchun 130025, China; yzg@mail.jlu.edu.cn

**Keywords:** piezoelectric pump, fluid cavity, piezoelectric vibrator, energy loss

## Abstract

The large load loss of piezoelectricity pumps leads to fluid energy in the fluid chamber during fluid transportation. In this paper, the output performance of a piezoelectricity pump is improved by changing the structure parameters of the fluid chamber to reduce the fluid load. The mechanism of fluid flow energy loss in the body cavity of hydraulic pumps is simulated and analyzed, and the influence of the dimensions of the inlet and outlet valves and the height of the cavity on fluid energy loss is obtained. The flow rate and pressure of inlet and outlet valves with different cavity heights and different driving frequencies are obtained. The results show that the flow rate and output pressure of the hydraulic pump are optimized when the cavity height is 3 mm, and the inlet and outlet valve diameters are 2.5 mm.

## 1. Introduction

Piezoelectric pumps have found wide application in the field of fluid delivery thanks to their simple structure, small volume, low manufacturing cost, and accurate output [1,2,3]. They have been widely used in areas such as drug delivery [4,5], electronic cooling [6,7], fuel cells [8,9], biomedicine [10,11], air compression and circulation [12], robotics [13], spray ink printers [14], microfluidic droplet generation [15], medical micro-sprayers [16,17], and medical applications [18,19]. Since the output displacement of the piezoelectric vibrator is small, the driving force generated will lead to the piezoelectric pump generating a small, low-pressure output flow. This is mainly due to serious energy loss of the fluid in the fluid cavity, the flow resistance, and the energy of the conveyed fluid in the pipeline. Such losses are unavoidable and can result in piezoelectric pumps not meeting the needs of practical applications. Therefore, it is necessary to modify the design of existing piezoelectric pumps to improve their performance. To this end, researchers are actively seeking to optimize the structure of the piezoelectric pump to improve the output performance of the piezoelectric pump by studying the pump cavity and other structural parameters that affect the performance of piezoelectric pumps [20,21].

To improve the output pressure and flow rate of the piezoelectric pump, Moradi-Dastjerdi et al. [22] used a passive polymer film to make a new type of flexible diaphragm and analyzed the influence of the structural parameters of their novel diaphragm on the performance of the piezoelectric pump. Their results showed that the thickness of the diaphragm plays an important role in the dynamic response of piezoelectric pump designs. Sathiya et al. [23] developed a piezoelectrically driven liquid pump with a two-degrees-of-freedom (2-DOF) cantilever resonator to improve the fluidity of liquids with different viscosities and studied the effect of the distance between the drives and the tube diameter on flow rate. Their results showed that the designed piezoelectric pump could accommodate liquids with different viscosities ranging from 0.23 mPas to 170 mPas, and its flow rate is higher than that of the single-degree-of-freedom (1-DOF) cantilever resonator piezoelectric pump. Nayak et al. [24] stated that piezoelectric pumps could provide larger pump flow under high-frequency conditions but, to a certain extent, limit the application of piezoelectric pumps to transport higher viscosity fluids. To combat this, the authors developed an amplification mechanism to increase the displacement of the piezoelectric actuator, and a piezoelectric hydraulic pump with a piezoelectric actuator (FAPA) was developed. Their experimental results showed that the displacement peak value of the piezoelectric pump at a frequency of 1 Hz is 369 mm, the lag is 16.22 mm, the flow rate is 25.36 mL/min, and the theoretical error is 0.75%.

To improve the oil supply pressure and flow of piezoelectric pumps, Wu [25] and others from Nanjing University of Aeronautics and Astronautics studied a valve distribution piezoelectric pump with a hydraulic amplification mechanism. Their experiments showed that the rebound of the valve core of the distribution valve would cause instantaneous flow fluctuation, and the diameter of the one-way valve core spring and valve seat can easily affect the fluctuation in fluid flow caused by this collision and rebound. The attenuator they used was able to reduce the fluctuations in the fluid flow by more than half. Based on the working principles of valveless piezoelectric pumps and the theory of hydraulic–electrical analogy, Zhang [26] of Guangzhou University and others established an equivalent circuit model of a valveless piezoelectric pump and compared the experimental and simulation results. Their study showed that when the driving voltage is 100 V, and the frequency is 6 Hz, the maximum flow rate of the valveless piezoelectric pump is 1.16 mL/min, indicating that the model has a good prediction for the calculation of the maximum output flow rate and the best working frequency of valveless piezoelectric pumps. Liu of Jilin University and others [27] designed multi-chamber series and parallel piezoelectric pumps successively by studying the influence of the number of pump chambers on the output performance of piezoelectric pumps. They determined that the frequency changes linearly, and the maximum output gas volume can reach 3600 mL/min.

To reduce the energy and load losses of piezoelectric pumps and improve their comprehensive performance, a piezoelectric pump with a new structure of compressible cavity is designed in this paper. A deformable elastic cavity is set inside each of the inlet and outlet valve pipelines of the piezoelectric pump to limit the flow of liquid outside the valve into the compressible cavity close to the valve to avoid load loss due to excessive liquid mass during the working process of the piezoelectric pump. The vibration damping of the piezoelectric pump system is also greatly reduced, and the energy conversion efficiency of the piezoelectric vibrator is improved at the same time.

## 2. Theoretical Analysis of Piezoelectric Pump Energy Loss

As fluid flows through a piezoelectric pump, flow resistance is generated between the various flow layers. During the working process, part of the mechanical energy of the piezoelectric vibrator is irreversibly converted into thermal energy generated by the flow resistance. The main source of energy loss within the piezoelectric pump is the fluid cavity, and the pipeline material, length, effective diameter, height, fluid viscosity, and fluid flow rate are the key factors affecting its performance.

The flow loss is given by [28]:(1)$$h_{f}=\mu \frac{l}{d}\frac{v^{2}}{2g}$$
where $h_{f}$ is the loss along the way, l is the length of the pipeline, d is the effective diameter of the pipeline, v is the average flow velocity of the section, g is the acceleration of gravity, and μ is the resistance coefficient along the way.

The performance of the piezoelectric pump is affected both by the energy loss caused by the friction between the fluid layers and by losses caused by the sharp change of the local boundary. The geometric loss is given by:(2)$$h_{j}=\zeta \frac{v^{2}}{2g}$$
where v is the average flow velocity of a section, g is the acceleration due to gravity, and ζ is the local loss coefficient.

It can be seen from the above analysis that the performance of the piezoelectric pump is closely related to the shape, length, and internal structure of the fluid cavity. Reducing the energy loss of the piezoelectric pump can directly improve the energy conversion efficiency of the piezoelectric vibrator and effectively improve the output performance of the pump.

## 3. Working Principle and Theoretical Analysis of a Compressible Cavity

During the working process of the piezoelectric pump, the fluid inside and outside the pump cavity becomes connected when the valve plate is opened, such that the drive of the piezoelectric pump is not limited to the fluid in the pump cavity, but also includes the fluid outside the valve. If the cavity structure is complex, the contact between the fluid and the cavity wall is high, and the cavity is narrow, the energy required by the piezoelectric vibrator to drive the fluid of the entire system is very large. As Figure 1, the schematic diagram of fluid flow in the fluid chamber, shows, the flow of fluid into the compressible cavity from the inlet tube can be regarded as a sudden expansion flow in the tube flow.

If we let the flow velocity in the inlet be $v_{1}$, the horizontal velocity component of the flow velocity in the compressible cavity be $v_{2}$, and the flow velocity at the outlet be $v_{3}$, then the formula for the head loss, *h_x_*, of sudden expansion pipe flow in the energy loss of piezoelectric pump is:(3)$$h_{x}=\frac{{\left(v_{1}-v_{2}\right)}^{2}}{2g}.$$

By the continuity equation:(4)$$\left(\frac{\pi }{4}\right){D_{1}}^{2}v_{1}=\left(\frac{\pi }{4}\right){D_{2}}^{2}v_{2}$$
where $D_{1}^{}$ is the effective diameter of the inlet pipe, $D_{2}$ is the equivalent diameter of the compressible cavity, which can be equivalent to $D_{2}=l$, then:(5)$${\left(v_{1}-v_{2}\right)}^{2}=\left(1-\frac{{D_{1}}^{2}}{{D_{2}}^{2}}\right){v_{1}}^{2}=\left(1-\frac{{D_{1}}^{2}}{l^{2}}\right){v_{1}}^{2}$$

Substituting Equation (5) into Equation (3) gives:(6)$$h_{x}=\left(1-\frac{{D_{1}}^{2}}{l^{2}}\right)\frac{{v_{1}}^{2}}{2g}$$

It can be seen from the above theoretical analysis that, as the height of the fluid cavity increases, the energy loss of the fluid flowing in the cavity also increases, resulting in a decrease in the output performance of the piezoelectric pump.

## 4. Fluid Flow Simulation Analysis

To simulate the fluid flow at different heights, we will define the upper part of the compressible cavity model as the inlet, with a pressure of 15 KPa, and define the side of the model as the inlet and outlet, with a pressure of 0. In addition, the elastic diaphragm material is defined as silicone rubber, and the fluid is defined as incompressible pure water. Fluid chamber heights of 2 mm, 3 mm, 4 mm, and 5 mm were simulated (Figure 2a–d, respectively).

The simulation results showed that when the 2 mm compressible cavity was used at the inlet and an incompressible cavity at the outlet, the maximum flow output increased by 7% (Figure 3).

The conclusions drawn from this simulation study were that: (1) when changing the height of the fluid chamber with constant inlet and outlet pressure, the flow at the outlet of the 2 mm compressible chamber and the incompressible chamber decreases with the increase of the height of the fluid chamber, (2) the flow output of the piezoelectric pump with a compressible cavity is significantly higher than that of a piezoelectric pump without a compressible cavity; and (3) the lower the height of the compressible cavity, the greater the flow output of the piezoelectric pump.

## 5. Experimental Studies

The layout of the prototype pump and the as-constructed device are shown in Figure 4 and Figure 5, respectively. The main components of the prototype pump are the pump body, the wheel valve plate, the valve plate cover, the piezoelectric vibrator, the vibrator diaphragm, and the metal wire welded on the vibrator. The instruments, components, and materials used in the experimental set-up are shown in Figure 5. The principal elements are an SP1641B type function signal generator, an HPV series piezoelectric ceramic driving power supply (with a driving voltage and waveform of 150 V sawtooth), and a Yilong Technology HH-2 type constant temperature water bath. Purified water, a fixed vise, a precision balance, a pressure gauge, a beaker, and a timer complete the set-up.

The material chosen for the compressible cavity diaphragms was silicone, as silicone membranes have strong corrosion resistance, and, as no plasticizer is added during the preparation process, they can maintain their elasticity for a long time, giving them a long service life. Diaphragms of thicknesses of 0.1 mm, 0.4 mm, 0.7 mm, and 1 mm were used. The dimensions of the pump are inlet and outlet valves Φ 7 × 0.03 (mm), inlet and outlet pipes Φ 1.2 × 1 × 100 (mm), valve cover plate Φ 7 × 0.05, and piezoelectric ceramic chip Φ 18 × 0.2 (mm), Φ 35 × 0.3 (mm).

We measured the flow rate at different frequencies without the compressible cavity and with compressible cavities with heights of 2 mm, 3 mm, 4 mm, and 5 mm. The resulting flow rate curves (Figure 6) show that the results of the experimental test are consistent with the results of the simulation analysis. The piezoelectric pump with a compressible cavity has a significantly higher flow output than the piezoelectric pump without a compressible cavity, and the lower the cavity height, the greater the flow output of the piezoelectric pump. The maximum flow output is with a 2 mm compressible cavity, and this is 30% higher than the maximum flow output without a compressible cavity.

We also measured the output pressure at different frequencies without the compressible cavity and with compressible cavities with heights of 2 mm, 3 mm, 4 mm, and 5 mm. The resulting curves (Figure 7) show that there is no significant change in the output pressure with or without the compressible cavity. At first, the flow rate of the piezoelectric pump without the compressible cavity gradually increases with increasing frequency. However, once the frequency reaches a certain value, the flow begins to decrease significantly, and the flow curve has an obvious peak value. In contrast, while the flow rate of the piezoelectric pump with the compressible cavity also initially increases with increasing frequency, if the frequency increases rapidly, the flow will fluctuate within a given frequency range, while if the frequency increases slowly, the flow gradually decreases, and the peak value is not obvious.

## 6. Conclusions

A new approach to mitigating the energy loss of fluid motion in a piezoelectric pump pipeline based on the use of a compressible cavity has been developed. Using a fluid-structure interaction simulation combined with the theory of fluid flow characteristics and head loss in the piezoelectric pump, it is verified that the height of the compressible cavity affects the fluid flow in the compressible cavity. Compared with the traditional piezoelectric pump without a compressible cavity structure, the output flow of the piezoelectric pump with a compressible cavity is significantly improved under the conditions of the same volume and power consumption by up to 30%.

## Figures and Tables

**Figure 1 micromachines-13-01100-f001:**
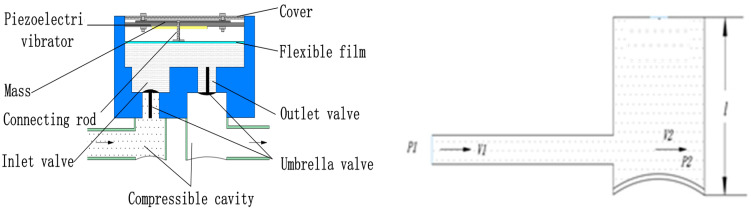
Schematic diagram of fluid flow in the compression chamber.

**Figure 2 micromachines-13-01100-f002:**
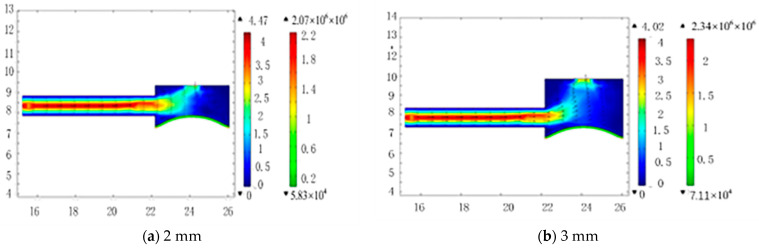
Dependence of fluid flow rate at the inlet of the compressible cavity on cavity height (**a**) 2 mm, (**b**) 3 mm, (**c**) 4 mm, and (**d**) 5 mm (Note: The left label in the legend is the fluid flow rate, the right is the height of the compression chamber).

**Figure 3 micromachines-13-01100-f003:**
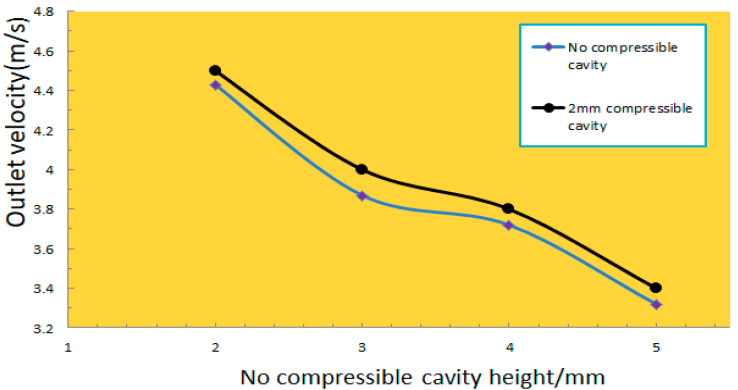
The flow rate curve at the outlet with a 2 mm compressible cavity at the inlet and an incompressible cavity at the outlet.

**Figure 4 micromachines-13-01100-f004:**
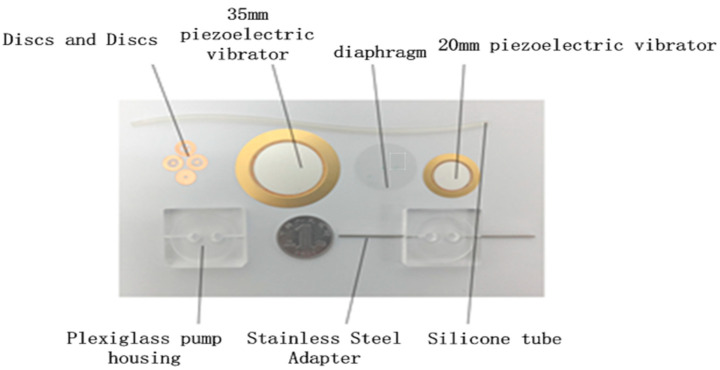
The layout of the prototype.

**Figure 5 micromachines-13-01100-f005:**
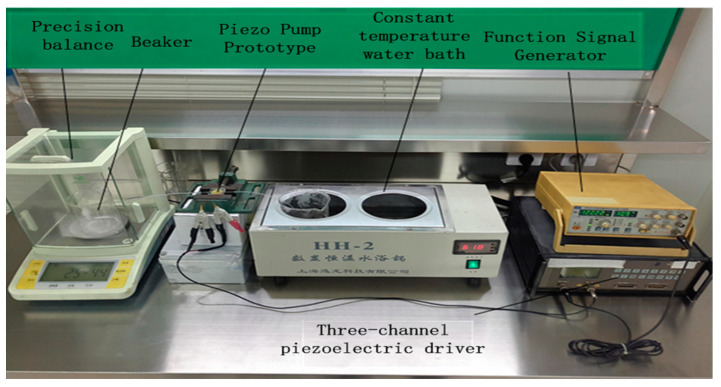
As-constructed device.

**Figure 6 micromachines-13-01100-f006:**
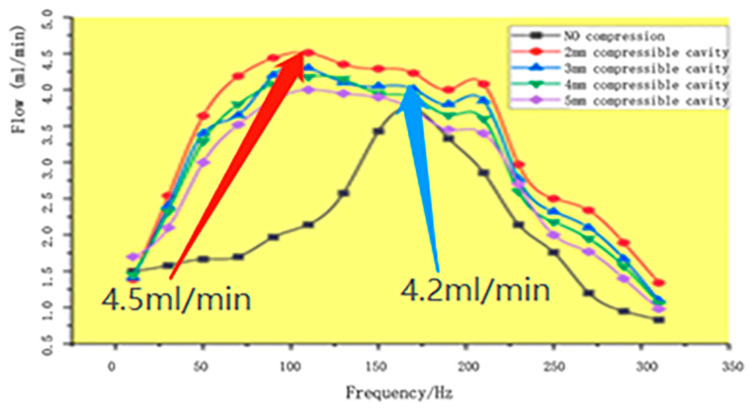
Frequency–flow rate comparison curves.

**Figure 7 micromachines-13-01100-f007:**
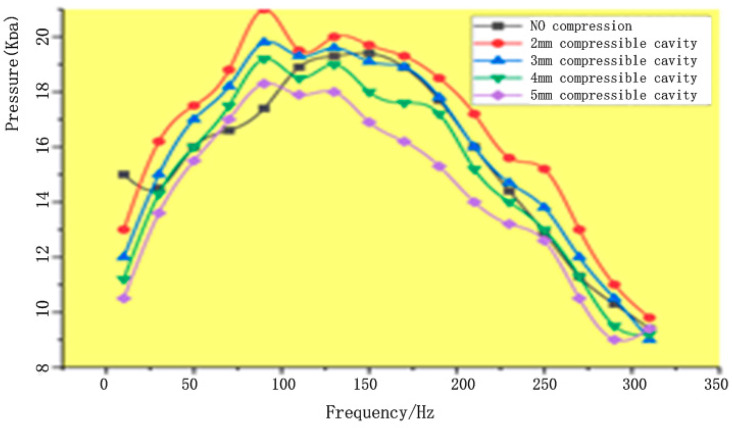
Frequency–pressure contrast curves.

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
