# Peer review of "Simulation Analysis and Performance Test of a Compressible Piezoelectric Pump Fluid Cavity"

_micromachines, 2022, doi:10.3390/mi13071100_

Round 1

Reviewer 1 Report

The authors designed a piezoelectric pump fluid chamber to improve pump performance. They performed flow simulations to analyze the fluid chamber design and also performed experiments to evaluate the proposed chamber design. Overall, the work seems premature and gives little meaning to this area. Description of simulation and experimental details is not presented properly. Why and how would the performance of the proposed system be improved by 30%? There is no reasonable and sufficient explanation for this. Therefore, I cannot recommend the publication of this work. Here are some more detailed comments.

  1. What is the rationale for eqns 1,2? Please provide relevant references or brief explanations.

  1. Include ‘No compressible cavity case’ in Fig. 6

  1. Experimental setup and analysis for the 'no-compression case' in Figure 9 is required. -> Why is there a 30% increase in compressible cavity cases? What mechanism?

Minor comments

  1. Typos

- p.6 line 165, Figure 10 -> check! Figure 10 is a Frequency-Pressure Contrast Curve

Reviewer 2 Report

Figure 2 – 5 can be combined into a single figure. There are two scale bars for each one of these figures, but it is not clear what they are presenting and what the unit is.

Figure 6: the font of the axis titles should be the same. Also, the titles are not properly placed in the middle of each axis.

The authors have claimed that the experimental results are in line with the simulation results. However, the validation is not clear. Please quantify your claim.

The results are not discussed in detail.

Reviewer 3 Report

Please give a detailed and clear physical drawing of the prototype.
